# Metabolite Alterations in Autoimmune Diseases: A Systematic Review of Metabolomics Studies

**DOI:** 10.3390/metabo13090987

**Published:** 2023-09-01

**Authors:** Abdulrahman Mujalli, Wesam F. Farrash, Kawthar S. Alghamdi, Ahmad A. Obaid

**Affiliations:** 1Department of Laboratory Medicine, Faculty of Applied Medical Sciences, Umm Al-Qura University, Makkah 24381, Saudi Arabia; wffarrash@uqu.edu.sa (W.F.F.); aaobaid@uqu.edu.sa (A.A.O.); 2Department of Biology, College of Science, University of Hafr Al Batin, Hafar Al-Batin 39511, Saudi Arabia; ksalghamdi@uhb.edu.sa

**Keywords:** autoimmune disease, metabolomics analysis, metabolites, biomarker and diagnosis

## Abstract

Autoimmune diseases, characterized by the immune system’s loss of self-tolerance, lack definitive diagnostic tests, necessitating the search for reliable biomarkers. This systematic review aims to identify common metabolite changes across multiple autoimmune diseases. Following PRISMA guidelines, we conducted a systematic literature review by searching MEDLINE, ScienceDirect, Google Scholar, PubMed, and Scopus (Elsevier) using keywords “Metabolomics”, “Autoimmune diseases”, and “Metabolic changes”. Articles published in English up to March 2023 were included without a specific start date filter. Among 257 studies searched, 88 full-text articles met the inclusion criteria. The included articles were categorized based on analyzed biological fluids: 33 on serum, 21 on plasma, 15 on feces, 7 on urine, and 12 on other biological fluids. Each study presented different metabolites with indications of up-regulation or down-regulation when available. The current study’s findings suggest that amino acid metabolism may serve as a diagnostic biomarker for autoimmune diseases, particularly in systemic lupus erythematosus (SLE), multiple sclerosis (MS), and Crohn’s disease (CD). While other metabolic alterations were reported, it implies that autoimmune disorders trigger multi-metabolite changes rather than singular alterations. These shifts could be consequential outcomes of autoimmune disorders, representing a more complex interplay. Further studies are needed to validate the metabolomics findings associated with autoimmune diseases.

## 1. Introduction

Autoimmune diseases include a wide range of clinical disorders, including rheumatoid arthritis (RA), multiple sclerosis (MS), inflammatory bowel diseases (IBDs), autoimmune liver diseases, and systemic lupus erythematosus (SLE), characterized by loss of self-tolerance by the immune system. Autoimmune diseases may be systemic or organ-specific, resulting in various complications and disabilities. Incidence of autoimmune disease varies owing to the diversity of diseases, affecting 5–10% of the population around the globe [1,2]. Several factors (genetic, environmental, and epigenetic factors) are involved in the development of autoimmune diseases [3]. Autoimmune diseases are poorly diagnosed owing to obscure symptoms and overlapping symptoms of various diseases. The majority of autoimmune diseases are multi-genic, with multiple susceptibility genes interacting to create the abnormal phenotype [4]. Several gene variants have been discovered for autoimmune diseases; however, their relationship with disease susceptibility remains elusive. Hence, a novel approach is required for comprehensive understanding of autoimmune disease biology, especially underlying molecular mechanisms and treatment strategies. Metabolomics is an emerging technology that has drawn the attention of the scientific community in order to identify disease biomarkers due to its cost-effectiveness, short time period for repeated measurements, and very close observation of the metabolic state of patients [5,6]. Metabolomics is employed to assess metabolites, which are the end products of biochemical processes, in both a quantitative and qualitative manner. Metabolomics provides better information about the status of metabolites that occur due to changes in gene expression. It is widely used in pharmaceutical industries and R&D for detecting biomarkers for diseases, identifying their signaling pathways, and assessing their efficacy. Metabolomics is classified into two categories: targeted metabolomics and untargeted metabolomics. Targeted metabolomics analyses specific metabolites, whereas non-targeted metabolomics is utilized to analyze the metabolites extracted from organisms systematically and comprehensively [7]. Metabolomics consists of various steps to identify novel disease biomarkers. Several biological specimens, including urine, cerebrospinal fluids, fecal extracts, serum, cyst fluid blisters, synovial fluids, plasma, seminal fluids, tissue extracts, dialysis fluids, exhaled breath condensates, bile fluids, and tissue biopsy extracts (aqueous and lipid), are the most common specimens utilized in metabolomics [8]. Analytical techniques, specifically mass spectroscopy (MS) in combination with various separation techniques (gas chromatography, liquid chromatography, HPLC, UPLC, and capillary electrophoresis) and nuclear magnetic resonance (NMR), are utilized for metabolomics studies [9,10,11]. Compared to NMR, MS is preferred for metabolomics as it requires small sample volumes and has high sensitivity and simple sample preparation [12]. pH is one of the major disadvantages of NMR, especially when dealing with urine samples. Several lines of evidence show the importance of metabolomics in the detection of various autoimmune diseases. Evidence from clinical trials has shown that metabolites can act as potential biomarkers for various diseases [13,14,15,16]. Previous clinical studies have reported that oncometabolites may act as diagnostic biomarkers for various carcinomas [17,18,19,20]. In addition to blood glucose, phospholipid profiling is also useful in identification of type 2 diabetes mellitus [21]. Trimethylamine N-oxide (TMAO) can also be used as a prognostic marker for patients with acute ischemic stroke who are at an increased risk of unfavorable clinical outcomes [22,23]. Another study reported a link between heart failure and urobilin and sphingomyelin (30:1) [24]. An association between carcinoma and eicosanoid metabolites was also reported [25]. For autoimmune diseases, serum, plasma, fecal extracts, urine, and other biological samples differ depending on the specific disease. Few studies have investigated biomarkers for diagnosis [26,27,28]. However, there is currently only specific tests for diagnosis for some autoimmune diseases. As a result, the current study aims to identify common metabolite changes across multiple autoimmune diseases.

## 2. Materials and Methods

### 2.1. Literature Search and Data Curation

The current systematic review was performed by following the guidelines of Preferred Reporting Items for Systematic Reviews and Meta-Analyses (PRISMA). MEDLINE, ScienceDirect, Google Scholar, PubMed, and Scopus (Elsevier) were searched for articles by using the following terms: “Metabolomics”, “Autoimmune diseases”, “Biological samples”, and “Metabolic changes”. Articles published in the English language up to March 2023 were included with no specific start date filter. Two hundred and fifty-seven articles were retrieved through a search strategy. After careful assessment of titles and abstracts, a total of 136 studies were un-contextualized for this study. Thus, 121 abstracts were left for further scanning. Then, 21 out of 121 studies were excluded due to repetition, and an additional 12 articles did not meet the predefined inclusion criteria, specifically those that were not published in English and lacked control groups. As a result, a final collection of 88 full-text articles were eligible for analysis. The selection process is depicted in Figure 1 using a PRISMA flow chart, outlining the study’s progression through these stages. The protocol of the study was registered with PROSPERO (registration number: CRD42023447059).

The initial screening of titles was conducted by the first reviewer (AM). Subsequently, two reviewers (AM and KSA) independently assessed the title and abstracts using an eligibility checklist to exclude irrelevant studies. Full texts of potentially eligible studies were retrieved for a comprehensive evaluation and final selection. Two reviewers (WFF and AAO) critically evaluated the quality and validity of the included studies. The first reviewer extracted the data, which were then verified by the second and third reviewers (KSA, AAO), and finally reviewed by the fourth reviewer (WFF) for accuracy and completeness. Consensus discussions were held to address any discrepancies and ensure study eligibility.

### 2.2. Data Synthesis

The outcomes of the included studies were summarized in tables mentioning the author, analytical technique, biological fluids, models, and the number of patients and controls. Metabolic changes with respect to different biological fluids were also summarized in tables.

### 2.3. Risk of Bias Assessment

For both the fluid samples and the studies, the risk of bias was assessed by using the AMSTAR 2 tool. We assessed the patient recruitment process and examined the information available/lack of information about the patients. Contrasting targeted and non-targeted metabolic analysis tactics were also evaluated and, finally, fluid sample collection techniques were also taken into consideration.

## 3. Results

The systematic search of different databases of published articles produced 88 studies. General characteristics of the included studies are shown in Table 1, which includes the author, biological fluids, analytical techniques, models, and sample sizes of different groups which allowed for further categorization of metabolomics changes in the biological fluids that were analyzed in the included studies: plasma, serum, feces, urine, and other biological fluids (synovial fluids, CSF, tears, peripheral blood monocytes, in vivo white matter, peripheral blood, and lymphocytes). In 12 out of 88 studies, other fluid samples were used in contrast to plasma, urine, feces, and serum [29,30,31,32,33,34,35,36,37,38,39,40].

### 3.1. Serum

A total of 33 studies assessed the metabolite changes in the serum of patients. In all studies, the analysis was performed on a human model. While changes were observed in various metabolites, not a single metabolite was found to be statistically significant across all the studies. In 11 studies, aromatic amino acids (tyrosine, tryptophan, and phenylalanine) were altered [44,45,49,51,53,57,79,80,105,110,111], four of which were found to be associated with SLE and PBC. Ten studies reported an alteration in branched amino acids (leucine, isoleucine and valine) [44,45,49,52,53,59,79,98,105,110], of which four studies found an association with SLE [49,52,53,59]. Alterations in fatty acids were observed in eight studies [44,50,53,57,61,104,106,108]. Among the eight studies, three were related to SLE [50,53,57]. The remaining metabolites that were shown to be significantly changed in the serum samples were linked to numerous metabolic pathways, including those related to lipid metabolism, ATP storage, nucleotide metabolism, oxidative stress, amino acid metabolism, and the TCA cycle (Table 2).

### 3.2. Plasma

A total of 21 studies assessed the metabolite changes in the plasma of patients. The analysis was conducted on human models in all studies. In all 21 plasma-based metabolite studies, not a single metabolite exhibited a consistent statistical significance across all experiments. Eleven out of twenty-one studies showed alterations in amino acid metabolism [41,46,62,68,69,70,72,91,96,100,101], of which five were shown to be associated with MS [62,68,69,70,72] and four with T1D [91,96,100,101]. Seven out of twenty-one studies showed an alteration in aromatic amino acids [41,46,62,68,70,96,101]. The alteration of metabolite levels in lipid metabolism has been reported in seven studies [41,56,65,66,93,99,101]. Of these, three and two studies were associated with MS [65,66] and T1D [93,99,101]. Fang et al. reported the alteration in membrane phosphoproteins and dihydroceramides [42]. Åkesson et al. reported the metabolic alteration in kynurenine pathways [55]. The remaining metabolites that were shown to be significantly changed in the plasma samples were linked to numerous metabolic pathways. These pathways included nucleotide metabolism, oxidative stress, amino acid metabolism, glycolytic metabolism, and the TCA cycle (Table 3).

### 3.3. Feces

A total of 15 studies assessed the metabolite changes in the feces of patients. The analysis was conducted on human models in all studies. Seven out of fifteen studies showed metabolic alterations in amino acid metabolism [58,74,75,77,81,85,88]. Of these, five studies were linked to CD [74,77,81,85,88]. Five out of fifteen studies showed an alteration in aromatic amino acids [58,74,75,78,81]. Of these, three studies were linked to CD [74,78,81]. Eight out of fifteen studies showed an alteration in bile acids [78,83,84,85,86,87,88,89]. Of these, five studies were linked to CD [78,84,85,86,88]. The remaining metabolites that were shown to be significantly changed in the fecal samples were linked to numerous metabolic pathways. These pathways included nucleotide metabolism, lipid metabolism, amino acid metabolism, and the TCA cycle (Table 4). Nonetheless, it is important to highlight that none of the identified metabolites exhibited consistent and significant alterations throughout all analyses.

### 3.4. Urine

A total of seven studies assessed the metabolite changes in the urine of patients. The analysis was conducted on human models in all studies. However, none of the identified metabolites showed consistent significant alterations across all studies. Four out of seven studies showed metabolic alterations in amino acid metabolism [54,71,92,94]. Of these, two studies were linked to T1D [92,94]. Three out of seven studies showed metabolic alterations in aromatic amino acids, especially tryptophan [54,71,94]. One of the studies reported a decrease in trigonelline and hippurate [111]. Deja et al. observed an increase in urea [92]. Another study reported an increase in bile acids [109]. The remaining metabolites that were shown to be significantly changed in the urine samples were linked to numerous metabolic pathways. These pathways included lipid metabolism, amino acid metabolism, and the TCA cycle (Table 5).

### 3.5. Other Biological Fluids

A total of 12 studies assessed the metabolite changes in the other biological fluids (synovial fluids, CSF, tears, peripheral blood monocytes, in vivo white matter, and peripheral blood and lymphocytes) of patients. The analysis was conducted on human models in all studies. In all 12 other biological fluid-based metabolite studies, not a single metabolite showed significant changes that were consistent across all experiments. Six out of twelve studies performed metabolomics analysis on CSF [32,34,35,36,38,39]. Two studies performed metabolomics analysis on synovial fluid [29,30]. Two studies performed metabolomics analysis on peripheral blood lymphocytes and monocytes [31,40]. One out of twelve studies performed metabolomics analysis on tears [37]. One study carried out metabolomics analysis on in vivo white matter [33]. Among the twelve studies, six reported metabolic changes in amino acid metabolism [30,31,33,37,38,39]. Of these, three studies were linked to MS [37,38,39]. A study conducted on blood samples from patients with MS observed a decrease in glucose and lactate levels [40]. There was another study conducted on CSF specimens from patients with MS, which observed a decrease in glycine, dimethylarginine, and glycerophospholipid PC-O (34:0), as well as hexoses [39]. Podlecka-Piętowska et al. analyzed the metabolic alteration in CSF from MS patients and observed a decrease in acetone, choline, urea, 1,3-dimethylurate, creatinine, isoleucine, myo-inositol, leucine, 3-OH butyrate, and acetyl-CoA [38]. A study conducted by Cicalini et al. reported an increase in amino acids and acylcarnitines in the tears of MS patients [37]. A study performed by Herman et al. reported a decrease in 3-methoxytyramine and caffeine in the CSF of MS patients [36]. Pieragostino et al. reported a decrease in phosphatydic acid and an increase in phosphatidylcholine and phosphatidylinositol in patients with MS [35]. Vingara et al. analyzed the metabolic alteration in in vivo white matter and reported a decrease in lipid metabolism in patients with MS [33]. Gonzalo et al. analyzed the metabolic alteration in CSF and reported a decrease in PPARϒ and an increase in 8-iso-prostaglandin F2α in patients with MS [32]. The remaining metabolites that were shown to be significantly changed in the other biological fluid samples were linked to numerous metabolic pathways. These pathways included nucleotide metabolism, lipid metabolism, amino acid metabolism, glycolytic metabolism, and the TCA cycle (Table 6).

## 4. Discussion

The metabolomics approach is a continuously evolving approach in the field of “omics” technology that offers a molecular view of disease pathophysiology and identifies disease biomarkers. Metabolomics also provides early diagnosis of diseases, better intervention, and monitoring of the progression of disease and the potency of treatment. The term autoimmune disease refers to a group of chronic disorders that are associated with a variety of metabolic changes that vary with the disease type. Given the absence of definitive cures for autoimmune diseases, patients are confronted with enduring illness and ongoing treatment throughout their lives. Hence, early diagnosis and recognition of various autoimmune diseases are essential to lessen disease progression and prevent painful conditions as well as co-morbidity and mortality caused by autoimmune diseases. The studies included in this systematic review analyzed the metabolic changes in various autoimmune diseases (rheumatoid arthritis, multiple sclerosis, systemic lupus erythematosus, Crohn’s disease, primary sclerosing cholangitis, primary biliary cholangitis, inflammatory bowel disease, ulcerative colitis, and type 1 diabetes) in serum, plasma, feces, urine, and other biological fluids including synovial fluids, CSF, tears, in vivo white matter, and peripheral blood monocytes and lymphocytes. All studies were carried out on patients. Mass spectroscopy and nuclear magnetic resonance were used in these studies. In most of the studies, mass spectroscopy was utilized in combination with various separation techniques. Metabolites that are identified through metabolomics analysis of various biological fluids are either reported as increased or decreased in contrast to controls. Various metabolites were found to increase or decrease, belonging to various metabolic pathways including TCA, glycolytic, amino acid metabolism, ATP metabolism, nucleotide metabolism, oxidative stress, lipid metabolism, and carbohydrate metabolism. A relatively consistent change in the proportion of metabolites was observed. However, there were instances of variation between individual cases. For instance, one of the studies reported an increase in the level of phosphatidylcholine in CSF specimens [35], while some studies observed a decrease in the level of phosphatidylcholine in tears and plasma [37,90]. We observed similar findings for other metabolites. It is possible that this may be due to interspecies differences in the metabolic process of patients, suggesting that further studies are required about pathophysiology and metabolomics. Further, for metabolomics findings to be applicable across species, it is imperative to identify both similarities and differences between animals and humans. Additionally, human clinical populations must be evaluated in order to confirm the utility of identified biomarker candidates in animal models. Different studies have reported metabolic changes associated with various autoimmune diseases. The metabolism of acylcarnitine and carnitine, changes in fatty acid metabolism, as well as TCA cycle metabolites have been linked to mitochondrial dysfunction [26,57,61,91,109,111,113]. Reactive oxygen species, antioxidant metabolites, glucogenic amino acid metabolites [58,114,115], and the accumulation of signaling metabolites were also reported [116]. Developing metabolites associated with mitochondrial dysfunction may be a focus for future research. The metabolism of various amino acids and lipids has been found to be similar in a number of studies. However, an alteration in phosphorylcholine has only been reported in a limited number of studies. In many studies, altered amino acid metabolism and the ratio of aromatic to branched amino acids have been found to be diagnostic indicators of autoimmune diseases, particularly SLE, MS, and CD [58,117,118]. However, the metabolic changes in the level of amino acids across the studies were different. This suggests that further studies are required to validate the ratio of aromatic and branched amino acids as a diagnostic indicator of autoimmune diseases. Maintaining body homeostasis requires the synthesis and degradation of proteins. Amino acid metabolism plays a crucial role in this biochemical process, including regulation of the innate and adaptive immune systems [119,120]. The utilization of amino acid metabolism changes as diagnostic markers offers several compelling advantages [121]. Amino acids, being stable and easily measurable in biological fluids, present a feasible and practical option for clinical assessment. Their involvement in a wide array of metabolic pathways makes them valuable indicators of physiological changes. Several studies have shown that alteration in amino acid metabolism is linked with various disease conditions, including cardiovascular disease [122], cancer [123,124], and autoimmune diseases [100,101,102,119,120,121]. A case report has shown that serum levels of aspartic and glutamic acids are linked with the development of myasthenia gravis [125]. Reports conducted on dietary protein restriction have demonstrated that branched amino acids contribute to promoting metabolic health [126,127]. In the current study, there were changes in serum levels of aromatic amino acids in 11 studies and branched amino acids in 10. A significant alteration in amino acid metabolism was observed in 11 plasma reports. Seven studies reported significant alterations in amino acid metabolism in feces whereas four studies reported them in urine. The above findings indicated that branched amino acid metabolism may act as a diagnostic biomarker for autoimmune diseases, specifically SLE, CD, and MS. Altered amino acids in other biological fluids may be related to different stages or severity of autoimmune diseases. However, it is necessary to validate the method with a larger study sample before it can be applied to diagnostic practice, due to the multifactorial, heterogeneous, and complex nature of these diseases. Only 88 articles met the inclusion criteria for the current study. There are, however, several articles on metabolomics and autoimmune diseases that did not meet our inclusion criteria or did not appear in databases due to keywords or database limitations. Thus, these studies were not chosen for this systematic review. A study should identify and control for confounding factors (dietary habits, patient demographics, and concurrent medical conditions) since biological fluids, especially plasma, urine, and serum, all reflect systemic metabolism. These confounding factors may be involved in the metabolic alterations, indicating that statistical modeling is required for development of diagnostic biomarkers of autoimmune diseases.

## 5. Conclusions

The findings of the current study suggest that alterations in amino acid metabolism, particularly aromatic and branched amino acids, may serve as potential diagnostic biomarkers for autoimmune diseases such as SLE, MS, and CD. We also observed altered amino acid metabolism in various biological fluids including plasma, feces, urine, synovial fluids, CSF, tears, peripheral blood monocytes, in vivo white matter, peripheral blood, and lymphocytes. The study also emphasizes the complexity and heterogeneity of autoimmune disorders, since several other metabolic alterations have been reported. These alterations within various metabolic pathways were linked to energy metabolism, oxidative stress, lipid metabolism, and nucleotide metabolism, suggesting that these shifts are likely consequences of autoimmune disorders. However, biomarkers are changed owing to slight alterations in the experimental environment. Hence, metabolomics analyses must be carefully performed in the laboratory. While amino acid metabolism emerges as a promising diagnostic biomarker, the study emphasizes that further studies are required to validate the method with a larger study sample before it can be applied to diagnostic practice, due to the multifactorial, heterogeneous, and complex nature of these diseases. Researchers need to explore the correlation between the severity or stages of autoimmune disease and amino acid metabolism in different biological fluids. Furthermore, studies are required to evaluate the relationship between alterations in amino acid metabolism in various biological fluids and different autoimmune diseases. They are also required to investigate the potential therapeutic targets and conduct longitudinal studies to evaluate the efficacy of the identified biomarkers over time.

## Figures and Tables

**Figure 1 metabolites-13-00987-f001:**
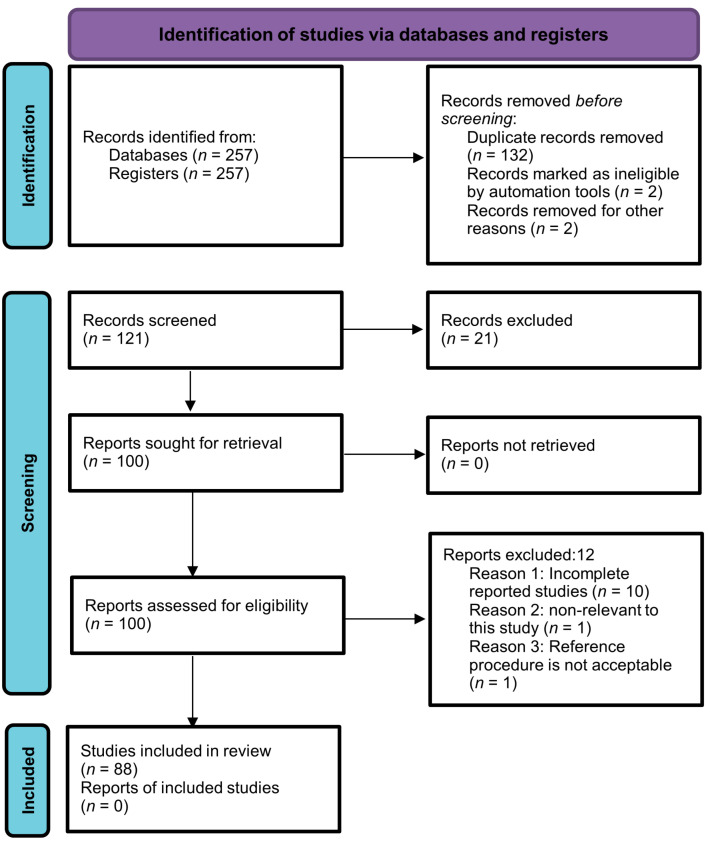
PRISMA flow chart illustrating the selection process of the studies.

**Table 1 metabolites-13-00987-t001:** General characteristics of the included studies.

S. No.	Author	Species	Fluid Sample	Analysis Technique	Sample Size
1	Madsen et al., 2011 [41]	Human	Plasma	GC-MS, UPLC-MS	RA = 20, HC = 10
2	Young et al., 2013 [30]	Human	Synovial fluid	GC-TOF MS	RA = 16, HC = 14
3	Yang et al., 2015 [29]	Human	Synovial fluid	GC-TOF MS	RA = 25, HC = 10
4	Fang et al., 2016 [42]	Human	Plasma	LC-MS	RA = 32, HC = 84
5	Zabek et al., 2016 [43]	Human	Serum	^1^H-NMR	RA = 20, HC = 30
6	Zhou et al., 2016 [44]	Human	Serum	GC-MS	RA = 33, HC = 32
7	Li et al., 2018 [45]	Human	Serum	UPLC-HRMS	RA = 30, HC = 32
8	Sasaki et al., 2019 [46]	Human	Plasma	CE-Q-TOFMS	RA = 49, HC = 10
9	Takahashi et al., 2019 [47]	Human	Serum	CE-TOF-MS	RA = 43, HC = 43
10	Hur et al., 2021 [48]	Human	Plasma	UPLC-MS/MS	RA = 128, HC = 12
11	Ouyang et al., 2011 [49]	Human	Serum	^1^H-NMR	SLE = 64, HC = 35
12	Wu et al., 2012 [50]	Human	Serum	GC-MS, LC-MS	SLE = 20, HC = 9
13	Perl et al., 2015 [31]	Human	Peripheral blood and lymphocytes	GC-MS, LC-MS	SLE = 36, HC = 39
14	Bengtsson et al., 2016 [51]	Human	Serum	GC-MS	SLE = 30, HC = 05
15	Guleria et al., 2016 [52]	Human	Serum	NMR	SLE = 22, HC = 30
16–17	Yan et al., 2016 [53,54]	Human	Urine and serum	GC-MS	SLE = 28, HC = 44
18	Åkesson et al., 2018 [55]	Human	Plasma	GC-MS, LC-MS, NMR	SLE = 132, HC = 30
19	Shin et al., 2018 [56]	Human	Plasma	GC-MS	SLE = 41, HC = 41
20	Li et al., 2019 [57]	Human	Serum	HPLC-MS	SLE = 17, HC = 17
21	Zhang et al., 2019 [58]	Human	Feces	UHPLC-MS	SLE = 32, HC = 26
22	Zhang et al., 2022 [59]	Human	Serum	UPLC-MS/MS	SLE = 52, HC = 21
23	Gonzalo et al., 2012 [32]	Human	CSF	LC-MS/UHPLC-MS	MS = 11, HC = 12
24	Mehrpour et al., 2013 [60]	Human	Serum	NMR	MS = 23, HC = 28
25	Vingara et al., 2013 [33]	Human	In vivo white matter	MRS with MRI	MS (RR) = 27, HC = 14
26	Dickens et al., 2014 [61]	Human	Serum	NMR	MS (RR) = 22, HC = 14
27	Reinke et al., 2014 [34]	Human	CSF	NMR	MS = 15, HC = 17
28	Pieragostino et al., 2015 [35]	Human	CSF	MALDI-TOF-MS, LC-MS/MS	MS(RR) = 12, HC = 13
29	Cocco et al., 2016 [62]	Human	Plasma	NMR	MS = 73, HC = 88
30	Gebregiworgis et al., 2016 [63]	Human	Urine	NMR	MS (RR) = 8, HC = 07
31	Lim et al., 2017 [64]	Human	Serum	UHPLC, GC-MS	MS (RR) = 50, HC = 49
32	Herman et al., 2018 [36]	Human	CSF	LC-MS/ELISA	MS (RR) = 30, HC = 10
33	Stoessel et al., 2018 [65]	Human	Plasma	LC-MS	MS (RR) = 10, HC = 63
34	Bhargava et al., 2019 [66]	Human	Plasma	GC-MS/LC-MS	MS = 18, HC = 18
35	Andersen et al., 2019 [67]	Human	Serum	2D GCxGC-TOFMS	MS = 12, HC = 13
36	Cicalini et al., 2019 [37]	Human	Tears	LC–MS/MS	MS = 12, HC = 21
37	Lorefice et al., 2019 [68]	Human	Plasma	NMR	MS = 21, HC = 21
38	Kasakin et al., 2019 [69]	Human	Plasma	LC–MS/MS	MS (RR) = 22, HC = 22
39	Podlecka-Piętowska et al., 2019 [38]	Human	CSF	NMR	MS = 19, HC = 19
40	Carlsson et al., 2020 [39]	Human	CSF	LC-HRMS, FIA-HRMS	MS = 12, HC = 12
41	Sylvestre et al., 2020 [70]	Human	Plasma	NMR	MS (RR) = 28, HC = 18
42	Gaetani et al., 2020 [71]	Human	Urine	HPLC–MS/MS	MS (RR) = 47, HC = 43
43–44	Zahoor et al., 2022 [40]	Human	Peripheral blood monocytes and serum	UPLC-MS/MS	MS (RR) = 35, HC = 14
45	Murgia et al., 2023 [72]	Human	Plasma	^1^H-NMR	MS = 42, HC = 22
46	De Preter et al., 2015 [73]	Human	Feces	GC-MS	CD = 83, HC = 16
47	Bjerrum et al., 2015 [74]	Human	Feces	^1^H-NMR	CD = 44, HC = 21
48	Lamas et al., 2016 [75]	Human	Feces	HPLC, LC-MS	IBD = 102, HC = 37
49	Coburn et al., 2016 [76]	Human	Serum	HPLC	UC = 137, HC = 38
50	Lee et al., 2017 [77]	Human	Feces	HRMS	CD = 31, UC = 22, HC = 19
51	Jacobs et al., 2016 [78]	Human	Feces	UPLC-MS	CD = 26, UC = 10, HC = 54
52–53	Kolho et al., 2017 [79]	Human	Serum and feces	UPLC-MS/MS	IBD = 69, HC = 29
54	Nikolaus et al., 2017 [80]	Human	Serum	HPLC	IBD = 291, HC = 291
55	Santoru et al., 2017 [81]	Human	Feces	^1^H-NMR, GC-MS, LC-QTOF-MS	CD = 50, UC = 82, HC = 51
56	Scoville et al., 2018 [82]	Human	Serum	HILIC/UPLC-MS/MS	CD = 20, UC = 20, HC = 20
57	Das et al., 2019 [83]	Human	Feces	LC-MS	IBD = 25, HC = 14
58	Weng et al., 2019 [84]	Human	Feces	GC-MS, LC-MS	CD = 172, UC = 107, HC = 42
59	Franzosa et al., 2019 [85]	Human	Feces	Untargeted LC-MS	CD = 68, UC = 53, HC = 34
60	Diederen et al., 2020 [86]	Human	Feces	^1^H-NMR, HPLC	CD = 43, HC = 15
61	Bushman et al., 2020 [87]	Human	Feces	UPLC-LC/MS	IBD = 28, HC = 37
62	Wang et al., 2021 [88]	Human	Feces	UPLC-MS/MS	CD = 29, HC = 20
63	Yang et al., 2021 [89]	Human	Feces	UPLC-MS/MS	UC = 32, HC = 23
64	Wu et al., 2022 [90]	Human	Plasma	UHPLC-HRMS	IBD = 30, HC = 15
65	Dutta et al., 2012 [91]	Human	Plasma	Untargeted UPLC-ToF MS	T1D = 07, HC = 07
66	Deja et al., 2013 [92]	Human	Urine	^1^H-NMR	T1D = 30, HC = 14
67	Balderas et al., 2013 [93]	Human	Plasma	LC-MS and CE-MS	T1D = 34, HC = 15
68	Galderisi et al., 2018 [94]	Human	Urine	LC-MS	T1D = 56, HC = 30
69	Frohnert et al., 2020 [95]	Human	Serum	LC-MRM/MS	T1D = 42, HC = 25
70	Lanza et al., 2010 [96]	Human	Plasma	^1^H-NMR, LC-MS	T1D = 09, HC = 09
71	Dutta et al., 2016 [97]	Human	Plasma	UPLC-TOF-MS	T1D = 14, HC = 14
72	Brugnara et al., 2012 [98]	Human	Serum	^1^H-NMR and GC-MS	T1D = 10, HC = 11
73	Knebel et al., 2016 [99]	Human	Plasma	GC-MS, LC-MS	T1D = 127, HC = 129
74	Lamichhane et al., 2019 [100]	Human	Plasma	GC-TOF-MS	T1D = 40, HC = 40
75	Bervoets et al., 2017 [101]	Human	Plasma	^1^H-NMR	T1D = 07, HC = 07
76	Zhang et al., 2022 [102]	Human	Serum	GC-TOF-MS	T1D = 76, HC = 65
77	Noso et al., 2023 [103]	Human	Serum	CE-FTMS, LC-TOF-MS	T1D = 23, HC = 03
78	Haukka et al., 2018 [104]	Human	Serum	UPLC-MS	T1D = 102, HC = 98
79	Wang et al., 2014 [105]	Human	Serum	^1^H-NMR	PBC = 41, HC = 14
80	Lian et al., 2015 [106]	Human	Serum	UPLC-MS	PBC = 20, HC = 25
81	Trottier et al., 2012 [107]	Human	Serum	LC-MS/MS	PBC = 12, PSC = 06, HC = 60
82	Bell et al., 2015 [108]	Human	Serum	UHPLC– MS/MS and GC– MS	PBC = 18, PSC = 21, HC = 10
83–84	Tang et al., 2015 [109]	Human	Serum and urine	UPLC/QTOF MS	PBC = 32, HC = 32
85	Hao et al., 2017 [110]	Human	Serum	^1^H-NMR	PBC = 29, HC = 41
86–87	Vignoli et al., 2018 [111]	Human	Serum and urine	^1^H-NMR	PBC = 20, HC = 19
88	Banales et al., 2019 [112]	Human	Serum	UHPLC-MS	PSC = 20, HC = 20

ELISA—Enzyme-linked immunosorbent assay, GC—Gas chromatography, LC—Liquid chromatography, MS—Mass spectroscopy, TOF—Time of flight, CE—Capillary electrophoresis, FTMS—Fourier transform mass spectroscopy, Q—Quadruple, HILIC—Hydrophilic interaction liquid chromatography, HRMS—High-resolution mass spectroscopy, MALDI—Matrix-assisted laser desorption/ionization, 2D GCxGC—Two-dimensional gas chromatography × gas chromatography, MRS—Magnetic resonance spectroscopy, MRI—Magnetic resonance imaging, TOFMS FIA—Flow injection analysis, MRM—Multiple reaction monitoring, UPLC—Ultra-pressure liquid chromatography, HPLC= High-pressure liquid chromatography, UHPLC—Ultra-high-pressure liquid chromatography, ^1^H-NMR—Proton nuclear magnetic resonance, HC—Healthy control, RA—Rheumatoid arthritis, SLE—Systemic lupus erythematosus, MS—Multiple sclerosis, T1D—Type 1 diabetes, CD—Crohn’s disease, PBS—Primary biliary cirrhosis, PSC—Primary sclerosing cholangitis, UC—Ulcerative colitis, IBD—Inflammatory bowel disease, RR—Remitting relapse.

**Table 2 metabolites-13-00987-t002:** Metabolite changes found in serum.

Author	Model	Metabolites/Metabolic Pathway
Zabek et al., 2016 [43]	Human	Up-regulated: 3-Hydroxyisobutyrate, acetate, NAC, acetoacetate, acetone Down-regulated: Isoleucine, lactate, alanine, creatinine, valine, histidine
Zhou et al., 2016 [44]	Human	Up-regulated: Docosahexaenoate, palmitelaidate, oleate, trans-9-octadecenoate, D-mannose, glycerol, ribose Down-regulated: 2-Ketoisocaproate, isoleucine, leucine, serine, phenylalanine, pyroglutamate, methionine, proline, threonine, valine, urate
Li et al., 2018 [45]	Human	Up-regulated: 4-Methoxyphenylacetic acid, glutamic acid, argininosuccinic acid, L-leucine, L-phenylalanine, L-tryptophan, L-proline, glyceraldehyde, fumaric acid, cholesterol Down-regulated: Capric acid, bilirubin
Takahashi et al., 2019 [47]	Human	Up-regulated: Betonicine, citric acid, quinic acid Down-regulated: Glycerol 3-phosphate, N-acetylalanine, hexanoic acid, taurine, 3-aminobutyric acid
Ouyang et al., 2011 [49]	Human	Up-regulated: Glucose, glycoprotein, lactate, VLDL, LDL Down-regulated: Valine, tyrosine, pyruvate, lysine, phenylalanine, HDL, cholesterol, isoleucine, histidine, alanine, phosphocholine, glycerol, glutamine, glutamate, creatinine, citrate
Wu et al., 2012 [50]	Human	Up-regulated: Medium-chain FA, 9-HODE, 13-HODE, LTB4, 5-HETE, gamma-glutamyl peptides Down-regulated: 1,2 Propanediol, 3-hydroxybutyrate, alpha ketoglutarate, citrate, G3P, lactate, malate, pyruvate, phosphocholine, essential polyunsaturated fatty acids (PUFAs), long-chain FA, acyl carnitines, GSH, methionine, cysteine, choline, pyridoxate, vitamin B6
Bengtsson et al., 2016 [51]	Human	Up-regulated: Urea, cystine, threonine, glucose Down-regulated: Lysine, fumaric acid, malic acid, methionine, tyrosine, alanine, asparagine, threonic acid, histidine, lactic acid, cysteine, citric acid, tryptophan
Guleria et al., 2016 [52]	Human	Up-regulated: Glucose and N-acetyl glycoprotein Down-regulated: Amino acids (leucine, valine, alanine, glycine, proline), citrate, choline, lactate
Yan et al., 2016 [53]	Human	Up-regulated: Methionine, glutamate, cystine, 1-monopalmitin, 1- monolinolein, 1-monoolein, 2-hydroxyisobutyrate Down-regulated: Amino acids (tryptophan, alanine, proline, glycine, serine, threonine, aspartate, glutamine, asparagine, lysine, histidine, tyrosine, valine, leucine, isoleucine), fructose, mannose, glucose, gluconic acidlactone, glycerol, oleic acid, arachidonic acid, fumarate, aminomalonate, threonate, alpha tocopherol
Li et al., 2019 [57]	Human	Up-regulated: Ceramides, phosphatidylethanolamine, ether phosphatidylcholine, diacylglycerol, sphingomyelin (SM), arachidonic acid, amino acids (arginine, L-glutamic acid, L-histidine), drug metabolites, 2-coumaric acid, acetylcholine, beta-guanidino propionic acid, xanthine, inosine, galacturonic acid, rac-glycerol 3 phosphate, trimethylamine N-oxide (TMAO) Down-regulated: Acylcarnitines, caffeine, hydrocortisone, itaconic acid, serotonin
Zhang et al., 2022 [59]	Human	Up-regulated: DG, SM, 1,5-anhydro-4-deoxy-D-glycero-hex-3-en-2-ulose, 8-(4-methoxy-2,3,6-trimethyl-phenyl)-6-methyl-octa-3,5-dien-2-one, Cer-BDS, phenylacetyl-L-glutamine, a-amino-g-cyanobutanoate, Pro-Leu, lysoDGTS, LDGTS, glycidyloleate Down-regulated: PE, 1-hexadecylthio-2-hexadecanoylamino-1,2-dideoxy-sn-glycero-3-phosphocholine, PC, Cer-NS, diisononyl phthalate, serylisoleucine, nervonic acid
Mehrpour et al., 2013 [60]	Human	Up-regulated: Glucose Down-regulated: Valine
Dickens et al., 2014 [61]	Human	Up-regulated: Fatty acids, beta-hydroxybutyrate Down-regulated: Glucose, phosphocholine,
Lim et al., 2017 [64]	Human	Up-regulated: Quinolinic acid Down-regulated: Kynurenic acid
Andersen et al., 2019 [67]	Human	Up-regulated: Pyroglutamate, laurate, acylcarnitine C14:1, N-methylmaleimide, phosphatidylcholines
Zahoor et al., 2022 [40]	Human	Down-regulated: Glucose, lactate
Coburn et al., 2016 [76]	Human	Up-regulated: L-citrulline (L-Cit), the L-Cit/L-Arg ratio Down-regulated: L-arginine
Kolho et al., 2017 [79] *	Human	Up-regulated in UC: Glycocholic acid, L-isoleucine, symmetric dimethylarginine, serine, phosphoethanolamine, proline, hexanoylcarnitine Up-regulated in CD: Neopterin, urea cycle, arginine and methionine metabolisms, namely L-arginine, dimethylglycine, asymmetric dimethylarginine, guanosine, L-octanoylcarnitine, betaine, L-cystathionine, citrulline, decanoylcarnitine Down-regulated: L-tryptophan, kynurenic acid, trimethylamine-N-oxide
Nikolaus et al., 2017 [80]	Human	Up-regulated: Quinolinic acid, Down-regulated: Tryptophan
Scoville et al., 2018 [82]	Human	Up-regulated: 54 metabolites in case of CD Down-regulated: 232 metabolites in case of CD and all decreased in case of UC
Frohnert et al., 2020 [95]	Human	Up-regulated: Serum glucose, ADP fibrinogen, mannose
Brugnara et al., 2012 [98]	Human	Up-regulated: Alanine and lactate, citrate, malate, fumarate, succinate Down-regulated: Valine, leucine
Zhang et al., 2022 [102]	Human	Up-regulated: TCA cycle metabolites (pyruvate, fuma indoleacetic acid rate, malate, linoleic acid), α-lactose, sorbitol, myo-inositol, sucrose, glycerol Down-regulated: 1,5-Anhydrosorbitol (1,5-anhydroglucitol), indoleacetic acid, d-mannose, d-galactose
Noso et al., 2023 [103]	Human	Up-regulated: 3-Phenylpropionic acid Down-regulated: Hypotaurine
Haukka et al., 2018 [104]	Human	Up-regulated: Carbohydrates, fatty acid, nucleotides, amino acids Down-regulated: γ-Glutamyl amino acids
Wang et al., 2014 [105]	Human	Up-regulated: Aromatic amino acids Down-regulated: Branched-chain amino acids
Lian et al., 2015 [106]	Human	Up-regulated: Bile acids Down-regulated: Free fatty acids, phosphatidylcholines, sphingomyelin, lysolecithins
Trottier et al., 2012 [107]	Human	Up-regulated: Total bile acids, taurine and glycine conjugates of primary bile acids in both PBC and PSC Down-regulated: Ratio of total glycine versus total taurine conjugates in case of PBC and secondary acids in case of PSC
Bell et al., 2015 [108]	Human	Up-regulated: Free fatty acid, acyl-carnitine, acetoacetate, BHBA Down-regulated: Lysolipids
Tang et al., 2015 [109]	Human	Up-regulated: Level of bile acid Down-regulated: Propionyl carnitine, butyryl carnitine
Hao et al., 2017 [110]	Human	Up-regulated: VLDL/LDL, taurine, glycine, phenylacetate, citrate, caprate, glycylproline, glucose, 3-hydroxyisovalerate, methionine, alanine Down-regulated: 4-Hydroxyproline, carnitine, 2-phosphoglycerate, citraconate, tyrosine, 3-hydroxyisobutyrate, inosine, thymidine, ornithine, tiglylglycine, urocanate, hippurate, n-acetylcysteine, isoleucine
Vignoli et al., 2018 [111]	Human	Up-regulated: Pyruvate, citrate, glutamate, glutamine, serine, tyrosine, phenylalanine, lactate
Banales et al., 2019 [112]	Human	Up-regulated: Glycholic acid, phosphatidylcholines Down-regulated: D(-)-2-aminobutyric acid

* Based on a partial least squares discriminant analysis (PLS-DA).

**Table 3 metabolites-13-00987-t003:** Metabolites changes found in plasma.

Author	Model	Metabolites/Metabolic Pathway
Madsen et al., 2011 [41]	Human	Up-regulated: Glyceric acid, D-ribofuranose, hypoxanthine Down-regulated: Histidine, threonic acid, methionine, cholesterol, asparagine, threonine
Fang et al., 2016 [42]	Human	Up-regulated: Lysophosphatidylinositol Down-regulated: Dihydroceramides, alkylphosphatidylethanolamine, alkenylphosphatidylethanolamines, phosphatidylserines
Sasaki et al., 2019 [46]	Human	Up-regulated: Tyrosine, phenylalanine Down-regulated: Lactate
Hur et al., 2021 [48]	Human	Up-regulated: Glucuronate, hypoxanthine
Åkesson et al., 2018 [55]	Human	Up-regulated: Kynurenine, quinolinic acid
Shin et al., 2018 [56]	Human	Up-regulated: Myristic, palmitoleic, oleic, and eicosanoic acid Down-regulated: Caproic, caprylic, linoleic, stearic, behenic, lignoceric, arachidonic, and hexacosanoic acid
Cocco et al., 2016 [62]	Human	Up-regulated: 3-OH-butyrate, acetoacetate, acetone, alanine, choline Down-regulated: Glucose, 5-OH-tryptophan, tryptophan
Stoessel et al., 2018 [65]	Human	Down-regulated: Glycerophospholipids, linoleic acid, lysoPC
Bhargava et al., 2019 [66]	Human	Up-regulated: Phospholipids, lysophospholipids, plasmalogen Down-regulated: Saturated and polyunsaturated fatty acids
Lorefice et al., 2019 [68]	Human	Up-regulated: Tryptophan Down-regulated: Acetoacetate, acetone, 3- hydroxybutyrate, glutamate, methylmalonate
Kasakin et al., 2019 [69]	Human	Up-regulated: Glutamate Down-regulated: Decenoylcarnitine, leucine–isoleucine
Sylvestre et al., 2020 [70]	Human	Down-regulated: Arginine, isoleucine, citrate, serine, phenylalanine, methionine, asparagine, histidine, myo-inositol
Murgia et al., 2023 [72]	Human	Up-regulated: Leucine Down-regulated: Circulating branched-chain AAs, valine, isoleucine
Wu et al., 2022 [90]	Human	Up-regulated: Phosphoethanolamine Down-regulated: Phosphotydilcholine
Dutta et al., 2012 [91]	Human	Up-regulated: Ketogenic and gluconeogenic amino acid, BCAA, glycerol, beta-hydroxybutyrate
Balderas et al., 2013 [93]	Human	Up-regulated: Free or non-esterified fatty acids, acetylarginine, hydroxytrimethyllysine, trimethyllysine Down-regulated: Tetrahydroaldosterone3-glucuronide
Lanza et al., 2010 [96]	Human	Up-regulated: Lactate, acetate, allantoin, ketones, leucine, isoleucine, valine, phenylalanine, tyrosine Down-regulated: Glycine, glutamate, threonine
Dutta et al., 2016 [97]	Human	Up-regulated: Carbohydrate metabolites: glucose, glucosamine, lactaldehyde, methylglyoxal, lactate, acetate, acetoacetate Down-regulated: Glycolytic metabolites such as pyruvate, dihydroxyacetone phosphate, TCA cycle metabolites
Knebel et al., 2016 [99]	Human	Up-regulated: PC species, biogenic amines, H1, AC C18:2, arachidonic acid levels Down-regulated: ᵹ-6-Desaturase (D6D), Val/Gly
Lamichhane et al., 2019 [100]	Human	Up-regulated: Methionine Down-regulated: Glutamic and aspartic acids
Bervoets et al., 2017 [101]	Human	Up-regulated: Glucose Down-regulated: Triglycerides, phospholipids and cho- linated phospholipids, serine, tryptophan, cysteine

**Table 4 metabolites-13-00987-t004:** Metabolites changes found in feces.

Author	Model	Metabolites/Metabolic Pathway
Zhang et al., 2019 [58]	Human	Up-regulated: Proline, L-tyrosine, L-methionine, L-asparagine, DL-pipecolinic acid, glycyl-L-proline, xanthurenic acid, kynurenic acid, L-carnosine, monoacylglycerol (MG) 22:6, MG 16:5, lysophosphatidylethanolamine (lysoPE) 16:0, lysophosphatidylcholine (lysoPC) 22:5, phosphatidylglycerol (PG) 27:2, 1,2-dioleoyl-rac-glycerol Down-regulated: Adenosine, adenosine 5′ -diphosphate (ADP), D-alaninyl-dalanine (D-Ala-D-Ala), lauryl diethanolamide, sulfoquinovosyl diacylglyceride (SQDG) 26:5, thiamine pyrophosphate, trigonelline, mucic acid
De Preter et al., 2015 [73]	Human	Up-regulated: 1-Ethyl3-methylbenzene, benzene acetaldehyde, phenol, 2-methyl propanal, carbon disulfide, 1-methoxy-4-methylbenzene Down-regulated: Pentanoate, hexanoate, heptanoate, octanoate, nonanoate
Bjerrum et al., 2015 [74]	Human	Up-regulated: Glycine, isoleucine, leucine, valine, alanine, tyrosine Down-regulated: Butyl, propyl
Lamas et al., 2016 [75]	Human	Down-regulated: Tryptophan, kynurenin
Lee et al., 2017 [77]	Human	Up-regulated: LysoPA Down-regulated: Pyridoxate
Jacobs et al., 2016 [78]	Human	Up-regulated: Bile acids, taurine, tryptophan, calprotectin
Kolho et al., 2017 [79] *		Up-regulated in UC: Aspartate, glycine, threonine, ornithine, creatinine, asparagine, glyceraldehyde, choline, kynurenine, histidine, taurine, phenylalanine, alanine, normetanephrine, allantoin, citrulline, carnosine, tryptophan, serine. None of the metabolites as significant as in CD Down-regulated in UC: CytosineDown-regulated in CD: Aspartate, threonine, asparagine, cytosine, histidine, taurine
Santoru et al., 2017 [81]	Human	Up-regulated: Biogenic amines, amino acids, lipids Down-regulated: B group vitamins
Das et al., 2019 [83]	Human	Up-regulated: Primary bile acids Down-regulated: Secondary bile acids
Wenig et al., 2019 [84]	Human	Down-regulated: Arachidic, oleic acid, ebacic acid, isocaproic acid, bile acids, riboflavin, nicotinate, pantothenate, 25-hydroxyvitamin D3
Franzosa et al., 2019 [85]	Human	Up-regulated: Sphingolipids, carboximidic acids, bile acids, cholesteryl esters, phosphatidylcholines, α-amino acids Down-regulated: Lactones, alkyl-phenylketones, ergosterols, quinolines, vitamin D, cholestrol
Diederen et al., 2020 [86]	Human	Up-regulated: Propionate, primary and conjugated bile acids Down-regulated: Secondary bile acids
Bushman et al., 2020 [87]	Human	Up-regulated: Calprotectin, cholate, chenodeoxycholate
Wang et al., 2021 [88]	Human	Up-regulated: Unconjugated bile acids, amino acids, including L-aspartic acid, linoleic acid, L-lactic acid Down-regulated: Conjugated bile acids
Yang et al., 2021 [89]	Human	Up-regulated: TGR5, taurocholic acid, cholic acid, taurochenodeoxycholate, glycochenodeoxycholate Down-regulated: VDR, secondary Bas, such as lithocholic acid, deoxycholic acid, glycodeoxycholic acid, glycolithocholic acid, taurolithocholate

* Based on a partial least squares discriminant analysis (PLS-DA).

**Table 5 metabolites-13-00987-t005:** Metabolites changes found in urine.

Author	Model	Metabolites/Metabolic Pathway
Yan et al., 2016 [54]	Human	Up-regulated: Valine, leucine, 3-hydroxyisobutyrate, fumarate, malate, cystine, pyroglutamarate, cysteine, threonate, uracil, pseudouridine, xanthine, urate, p-cresol, 2-hydroxyisobutyrate, tryptophan, glyceric acid, myo-inositol, 2,3-dihydroxybutyrate, 2,4-dihydroxybutyrate, 3,4- dihydroxybutyrate, 3,4,5-trihidroxypentanoic acid glutarate
Gebregiworgis et al., 2016 [63]	Human	Up-regulated: Trimethylamine N-oxide, 3-hydroxyisovalerate, hippurate, malonate Down-regulated: Creatinine, 3-hydroxybutyrate, methylmalonate
Gaetani et al., 2020 [71]	Human	Up-regulated: Indole-3-propionic acid Down-regulated: Urinary tryptophan, kynurenine, anthranilate, serotonin, K/T ratio
Deja et al., 2013 [92]	Human	Up-regulated: Urea Down-regulated: Pyruvate, citrate, succinate, glycine, phenylalanine, valine, alanine
Galderisi et al., 2018 [94]	Human	Up-regulated: Tryptophan, phenylalanine
Tang et al., 2015 [109]	Human	Up-regulated: Level of bile acid Down-regulated: Propionyl carnitine, butyryl carnitine
Vignoli et al., 2018 [111]	Human	Down-regulated: Trigonelline, hippurate

**Table 6 metabolites-13-00987-t006:** Metabolite changes found in other biological fluids.

Author	Model	Fluid	Metabolites/Metabolic Pathway
Young et al., 2013 [30]	Human	Synovial fluid	Up-regulated: 3-Hydroxybutyrate, lactate, acetylglycine, taurine, glucose Down-regulated: LDL-lipids, alanine, methylguanidine
Yang et al., 2015 [29]	Human	Synovial fluid	Up-regulated: Lactic acid, carnitine, diglycerol, pipecolinic acid, betamannosylglycerate Down-regulated: Valine, citric acid, gluconic lactone, glucose, glucose-1-phosphate, mannose, 5-methoxytryptamine, D-glucose, ribitol
Perl et al., 2015 [31]	Human	Peripheral blood and lymphocytes	Up-regulated: Kynurenine, methionine sulfoxide, cystine, OAA, PEP, DHAP, 3 PG, R5P, guanine, guanosine, GDP, dGDP, AMP, ADP, cytosine, dCTP, PHE Down-regulated: Cysteine, inosine
Gonzalo et al., 2012 [32]	Human	CSF	Up-regulated: 8-Iso-prostaglandin F2α Down-regulated: PPARϒ
Vingara et al., 2013 [33]	Human	In vivo white matter	Up-regulated: N-acetyl-aspartate, glutamate/glutamine, choline Down-regulated: Lipid
Reinke et al., 2014 [34]	Human	CSF	Up-regulated: Threonate, choline, myo-inositol Down-regulated: Phenylalanine, mannose, citrate, 3-hydroxybutyrate, 2-hydroxyisovalerate
Pieragostino et al., 2015 [35]	Human	CSF	Up-regulated: Phosphatidylcholine, phosphatidylinositol Down-regulated: Phosphatydic acid
Herman et al., 2018 [36]	Human	CSF	Up-regulated: Trigonelline, citrulline, O-succinyl-homoserine, N6-(delta2-isopentenyl)- adenine, pipecolate, 1-methyladenosine, 4-acetamidobutanoate, 5-hydroxytryptophan, kynurenate N-acetylserotonin Down-regulated: 3-Methoxytyramine, caffeine
Cicalini et al., 2019 [37]	Human	Tears	Up-regulated: Amino acids, acylcarnitines Down-regulated: Phosphotydilcholine, lyso-phosphotydilcholine sphingomyelins
Podlecka-Piętowska et al., 2019 [38]	Human	CSF	Down-regulated: Acetone, choline, urea, 1,3-dimethylurate, creatinine, isoleucine, myo-inositol, leucine, 3-OH butyrate, acetyl-CoA
Carlsson et al., 2020 [39]	Human	CSF	Up-regulated: Glycine, asymmetric dimethylarginine, glycerophospholipid PC-O (34:0), hexoses
Zahoor et al., 2022 [40]	Human	Peripheral blood monocytes and serum	Down-regulated: Glucose, lactate

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
