# Peer review of "Metabolite Alterations in Autoimmune Diseases: A Systematic Review of Metabolomics Studies"

_metabolites, 2023, doi:10.3390/metabo13090987_

Round 1

Reviewer 1 Report

Mujalli et al. reviewed the studies involving metabolomics studies and autoimmune diseaeses, and suggests that amino acid metabolism may serve as a diagnostic biomarker for autoimmune diseases. Below are my comments:

Major:

The authors reviewed 88 studies, and listed in tables the metabolite changes observed in different studies. However, a review should summarize the reported findings and provide insights for future directions, which is missing in the manuscript. A figure may better present such a summary.

Since many studies have showed the relationship between amino acid metabolism and autoimmune diseases, and hence suggest that amino acid metabolism can be the diagnostic biomarkers, what is the new finding from the review?

As other metabolites have also shown changes in autoimmune diseases, what are the advantages of using amino acid metabolism changes as diagnostic markers?

Minor:

Line84-86: the statement is confusing, please rewrite.

Line178-184: Authors listed changes in metabolites such as amino acids and fatty acids, which is in conflict with the statement “All 34 serum-based metabolite studies found no significant alterations in any single metabolite”.

Line197, 219, 236, 254: same problem as above.

Line290-291: please rewrite the sentence.

Line 306-308: the authors firstly mentioned the studies are “consistent”, however, later pointed out opposite observations from different studies.

Line317-329: please cite references for these statements.

Line356: please briefly describe the “findings of the current study”.

Line358-359: single metabolite alteration should not be the cause of autoimmune disorders; rather these alterations are more likely caused by the autoimmune disorders.

Line 360-361: please list references that support the statement.

Quality of English Language is good.

Reviewer 2 Report

In this review, the authors have attempted to identify common metabolites across different autoimmune diseases in this review. The authors suggest amino acid metabolism could serve as a diagnostic for different autoimmune diseases. 

Can the authors write a paragraph about the current understanding of metabolites and their relevance in clinical settings?

What is the new information provided in the review? 

Reviewer 3 Report

This systemic review aims to summary the published data regarding to metabolites alternation among autoimmune patients. It's an interesting topic, but I found several places the paper could be improved.

1. For the introduction part, the authors didn't include enough background, e.g. the mechanisms of metabolites changes in autoimmune patients compares to healthy population. Also, "gene polymorphism" is a different concept than "multi-genetic", the content in Line 38-39 and reference 4 is very awkward and inappropriate.  

2. Since the metabolomics includes "serum" and a variety of "fluids"(line 56-58), it's not "non-invasiveness "(line 45-46).

3. Line 85, ". Finally, 100 full-text articles were selected for review", should be "88"?

4. Horizontal and vertical rules need to be removed from the body of the tables. 

5. Results seems to be simply copying the reference articles, not much in the discussion section, either.  The separated "Amino acid metabolism" feels not properly organized to me. After reading the paper, I didn't get much insight on the topic.

Round 2

Reviewer 1 Report

Thanks for the efforts in addressing the concerns. As these responses are important for readers to understand the topic and related studies, they should be included in the revised manuscript, and listed in the response document.

Author Response

We wish to express our appreciation to the reviewers for their insightful comments, We sincerely appreciate your contribution to the refinement of our manuscript. We believe that the revised version of our paper addresses all concerns by the referees in detail. All modifications are highlighted in the manuscript

  

Sincerely,   
Abdulrahman Mujalli 
On behalf of all authors  

Round 2

Reviewer 1

Thanks for the efforts in addressing the concerns. As these responses are important for readers to understand the topic and related studies, they should be included in the revised manuscript, and listed in the response document.

Response: Thank you for your feedback. We completely agree with your suggestion. All modifications previously made are now included in the revised manuscript

Reviewer 3 Report

By reading through the modified manuscript, I noticed that the authors have made changes to most of my comments and did a pretty nice job. I'd only request further edition in Discussion section.

I understand that the authors wanted to emphasize the amino acids changes among other metabolites. However, the transition of the three paragraphs doesn't make a lot of sense. I suggest removing the subtitles in discussion, and rewording.

Author Response

(The authors gave the same response as above.)

Round 2

Reviewer 3

By reading through the modified manuscript, I noticed that the authors have made changes to most of my comments and did a pretty nice job. I'd only request further edition in Discussion section.

I understand that the authors wanted to emphasize the amino acids changes among other metabolites. However, the transition of the three paragraphs doesn't make a lot of sense. I suggest removing the subtitles in discussion, and rewording.

Response: Thank you for taking the time to review the revised manuscript. We appreciate your positive feedback on the changes made based on your previous comments. As suggested, we removed the subtitles from the three paragraphs and we reworded the discussion.

Round 3

Reviewer 1 Report

I don't have further suggestions for the manuscript.

Minor editing of English language required.